| Antimicrobial Chemotherapy | Research Article

# *In vitro* effects of the new oral β-lactamase inhibitor xeruborbactam in combination with oral β-lactams against clinical *Mycobacterium abscessus* isolates

Izumi Yamatani,[1,2] Akio Aono,[1] Keiji Fujiwara,[1] Takahiro Asami,[1] Keisuke Kamada,[1] Yuta Morishige,[1] Yuriko Igarashi,[1] Kinuyo Chikamatsu,[1] Yoshiro Murase,[1] Hiroyuki Yamada,[1] Akiko Takaki, Kosaku Komiya,[2] Satoshi Mitarai[1,3]

**ABSTRACT**   Non-tuberculosis mycobacteria (NTM), particularly *Mycobacterium abscessus* subsp. *abscessus* (*M. abscessus*), are increasingly being recognized as etiological agents of NTM pulmonary disease. However, treatment options for *M. abscessus* are limited owing to their natural resistance to most antibiotics, including β-lactams. *M. abscessus* produces a class A β-lactamase, whose activity is inhibited by cyclic boronic acid β-lactamase inhibitors. We aimed to evaluate the *in vitro* effects of xeruborbactam, a cyclic boronic acid β-lactamase inhibitor, against *M. abscessus* when combined with five β-lactams (amoxicillin, tebipenem, cefdinir, cefuroxime, and cefoxitin). The drug susceptibilities of 43 *M. abscessus* clinical isolates obtained from 43 patients between August 2005 and May 2014 were tested. The MIC results for each β-lactam with or without 4 µg/mL xeruborbactam were examined. Xeruborbactam lowered the $MIC_{90}$ values of tebipenem, amoxicillin, cefuroxime, and cefdinir by 5, ≥4, 3, and 3 dilutions, respectively. The $MIC_{90}$ values of cefoxitin without xeruborbactam were 32 µg/mL and did not change upon the addition of xeruborbactam. The lowest $MIC_{90}$ value was obtained for tebipenem with xeruborbactam. Almost all isolates had an MIC of 4 µg/mL; one isolate had an MIC of 2 µg/mL. With respect to the susceptibility to the same family drug, the number of susceptible isolates increased from 1/43 (2%) to 43/43 (100%) for tebipenem with xeruborbactam. Combining tebipenem and xeruborbactam could be considered an effective all-oral regimen that benefits outpatient treatment of *M. abscessus* pulmonary disease.

**IMPORTANCE**   *Mycobacterium abscessus* subsp. *abscessus* (*M. abscessus*) disease is treated in two phases; injectable drugs for initial followed by others for continuation. There is a need to develop all-oral treatment methods for *M. abscessus* infection, especially in the continuation phase. However, treatment options for *M. abscessus* are limited owing to their natural resistance to most antibiotics. This is the first report to evaluate the in vitro effects of xeruborbactam, a cyclic boronic acid β-lactamase inhibitor capable of inhibiting the class A β-lactamase produced by *M. abscessus*, against 43 *M. abscessus* clinical isolates when combined with five β-lactam antibiotics. Xeruborbactam lowered the MIC90 values of tebipenem by five dilutions, and the number of susceptible isolates increased from 1/43 (2%) to 43/43 (100%). We showed that the tebipenem-xeruborbactam combination might be of interest to explore further as a potentially effective oral regimen for outpatient treatment of *M. abscessus* pulmonary disease.

**KEYWORDS**   β-lactamase inhibitor, cyclic boronate acid β-lactamase inhibitor, β-lactam, *Mycobacterium abscessus*, xeruborbactam, tebipenem

Address correspondence to Izumi Yamatani, yamataniizumi7@gmail.com.

The authors declare no conflict of interest.

See the funding table on p. 9.

Non-tuberculosis mycobacteria (NTM) are normally present in natural and living environments, and the prevalence of NTM pulmonary disease has been increasing globally (1). Among these, the *Mycobacterium abscessus* complex (MABC) stands out as an emerging pathogen. In Asia, including Japan, it is the second most prevalent pathogen after the *Mycobacterium avium-intracellulare* complex (1, 2). MABC comprises three subspecies, i.e., *M. abscessus* subsp. *abscessus* (*M. abscessus*), *M. abscessus* subsp. *massiliense* (*M. massiliense*), and *M. abscessus* subsp. *bolletii* (*M. bolletii*). MABCs are rapidly growing mycobacteria and cause invasive pulmonary infections, particularly in patients with structural lung disorders, such as cystic fibrosis and bronchiectasis (1).

MABC differs in susceptibility to the key drug, macrolide, based on subspecies. Though most *M. abscessus* and *M. bolletii* are macrolide-resistant, *M. massiliense* is macrolide-susceptible. Two primary mechanisms underlie their macrolide resistance. One mechanism involves inducible resistance owing to the erythromycin ribosomal resistance methylase, *erm*(41) (3). Partial truncation of the *erm*(41) observed in *M. massiliense* causes macrolide susceptibility. The T28C sequence variant (sequevar) observed in *M. abscessus* fails to function in the *erm*(41) gene and confers macrolide susceptibility. In both subspecies, as with other NTM, mutations in the 23S rRNA gene *rrl* cause high macrolide resistance, regardless of the *erm*(41) activity (4).

The current recommendations suggest a two-phase treatment approach for MABC pulmonary disease. The initial phase involves a combination of injectable antibiotics, followed by a continuation phase primarily using oral or inhaled antibiotics (5). Although multidrug therapy is recommended for MABC pulmonary disease, the treatment failure rate is high (6), and treatment options are limited owing to the natural resistance of MABC to many antibiotics. In the continuation phase, clofazimine and linezolid are the only two oral antibiotics recommended for use, and their long-term use is complicated because of adverse events (7, 8). Owing to the long-term use of antibiotics in the continuation phase, the development of novel and highly effective all-oral anti-MABC treatments is urgently needed.

MABC produces the class A β-lactamase, Bla$_{Mab}$, which can hydrolyze penicillin, most cephalosporins, and carbapenems. Class A β-lactamases are not inhibited by traditional β-lactamase inhibitors, such as clavulanic acid, tazobactam, and sulbactam (9). In contrast, diazabicyclooctane β-lactamase and cyclic boronic acid β-lactamase inhibitors can inhibit Bla$_{Mab}$ (10–16). Combinatorial treatment involving β-lactams and these novel β-lactamase inhibitors against MABC has predominantly been studied *in vitro*. Xeruborbactam is a cyclic boronic acid β-lactamase inhibitor with inhibitory activity against major members of classes A, B, C, and D β-lactamases. It is available for intravenous or oral administration and has completed phase 1 clinical trials (ClinicalTrials.gov identifiers NCT04380207 and NCT04578873). Therefore, we hypothesized that combining β-lactams and xeruborbactam might restore the susceptibility of MABC to β-lactams.

MABC has two distinct colony morphotypes, which exist as smooth and rough colonies. A recent study reported that patients with rough MABC morphotypes had worse clinical outcomes than those with smooth morphotypes (17).

This study aimed to evaluate the *in vitro* effects of five clinically available β-lactams (amoxicillin, tebipenem, cefdinir, cefuroxime, and cefoxitin) against *M. abscessus* when combined with xeruborbactam. Furthermore, a secondary objective was to study the differences in the effects of the two morphotypes.

## MATERIALS AND METHODS

### Bacterial strains

Forty-three *M. abscessus* isolates obtained primarily from airway specimens of 43 patients at Fukujuji Hospital, Japan Anti-Tuberculosis Association, between August 2005 and May 2014 were investigated. *M. abscessus* was cultured on 2% Ogawa medium (Kyokuto Pharmaceutical Industrial, Tokyo, Japan), an egg-based medium mainly used in Japan and some Asian countries instead of the Löwenstein-Jensen medium, at 30°C. Species

and subspecies were identified based on multiplex PCR results and *rpoB* sequences according to previous reports (18, 19). The sequence of *erm*(41) was determined using the pyrosequencing method, as reported previously (20). All isolates were stored at −80°C at the Research Institute of Tuberculosis, Japan Anti-Tuberculosis Association, until use. As this study involved the use of only clinical isolates and no additional samples or personal information were obtained from the patients, ethics approval was not required.

## Antibiotics

Cefdinir (98% purity) and cefuroxime (100%) were obtained from Cayman Chemical (Ann Arbor, MI, USA), and amoxicillin (87.2%), tebipenem (100%), and cefoxitin (95.5%) were obtained from Sigma-Aldrich (St. Louis, MO, USA). (1R,2S)-Xeruborbactam (disodium; 95%) was obtained from Chem Scene LLC (Monmouth Junction, NJ, USA), and as per the manufacturer's instructions, it is soluble in dimethyl sulfoxide; there are no data available about the water solubility of the compound. We dissolved xeruborbactam and individual β-lactams in cation-adjusted Mueller-Hinton broth (CAMHB; pH 7.4) and stored the solutions at −80°C until use. When measuring the MIC values of individual β-lactams, a two-fold dilution series of the β-lactam stock solution was prepared using CAMHB. However, when measuring the MIC values of individual combinations of β-lactams and xeruborbactam, a two-fold dilution series of the β-lactam stock solution was prepared using CAMHB containing xeruborbactam.

Given this study's aim to explore an oral MABC regimen suitable for the continuation phase, we selected the commercially available oral β-lactams, amoxicillin, tebipenem, cefdinir, and cefuroxime, which were previously reported to be effective when combined with novel β-lactamase inhibitors (10, 11, 13–16, 21). Cefoxitin was already included as the treatment option in the ATS/ERS/ESCMID/IDSA clinical practice guidelines (5) and was used as a reference in this study. The MIC of xeruborbactam was not measured because novel β-lactamase inhibitors were previously demonstrated to lack antibacterial activity (13, 14).

## Antimicrobial susceptibility test

The MIC for each isolate was determined using the broth microdilution method with 96-well microtiter plates (Thermo Fisher Scientific, Waltham, MA, USA). The culture medium used was CAMHB (pH 7.4), as per the guidelines outlined in Clinical and Laboratory Standards Institute M24, third edition (22). Antibiotic solutions were prepared with a twofold dilution. The final drug concentrations were as follows: amoxicillin, tebipenem, cefdinir, and cefuroxime (0.25–256 µg/mL); cefoxitin (0.125–128 µg/mL). Xeruborbactam was used at a final concentration of 4 µg/mL, a level that can be easily attained in human plasma (23–25). The half-life of xeruborbactam is estimated to be approximately 30 h (24, 25), while that of avibactam is approximately 2 h (26). Although some β-lactams, such as imipenem, are unstable during long incubation time, a phenomenon that could potentially result in artificially high MIC values, avibactam is reported to be stable during incubation (14, 27). Therefore, we have considered that xeruborbactam was probably stable during incubation.

Bacterial colonies were transferred to test tubes with 5 mm diameter glass beads. Sterile distilled water was subsequently added, and the tubes were agitated using a vortex mixer to disperse any aggregated bacteria and ensure even suspension. Next, the bacterial suspension was resuspended in sterile distilled water and adjusted to a McFarland 0.5 standard by measuring absorbance at an optical density of 530 nm ($OD_{530}$). The final inoculum suspensions, each having a concentration of $5 \times 10^5$ CFU/mL, were prepared by transferring 90 µL of the 0.5 McFarland suspensions to 9 mL CAMHB. Subsequently, 50 µL of this final inoculum suspension was added to each well of a microtiter plate containing 50 µL of CAMHB with the dissolved antimicrobial agent, resulting in a final bacterial density of approximately $5 \times 10^4$ CFU in each well. Plates were incubated at 30°C under aerobic conditions. When the control strain grew sufficiently (on days 3–5), the MIC values for individual β-lactams were assessed. If

growth remained insufficient on day 5, incubation was extended to 14 days until sufficient growth was achieved.

The colony morphotype was decided based on appearance with the naked eye and sense of touch with the inoculating loop; the smooth morphotypes are characterized by shiny, smooth, and wet colonies, and the rough morphotypes are characterized by larger, markedly rugged, waxier, and dry colonies. The 96-well microtiter plates and colonies growing on 2% Ogawa medium were observed by more than two persons, and the final MIC values and colony morphotypes were determined by consensus.

Quality control (QC) was performed using *Mycobacterium peregrinum* ATCC 700686. The QC range was applied as recommended in CLSI M24S (28). Additionally, *M. abscessus* ATCC 19977 was used as an internal QC strain.

$MIC_{50}$ and $MIC_{90}$ were defined as the antibiotic concentrations preventing the growth of 50% and 90% of the isolates, respectively. The method for determining the $MIC_{50}$ and $MIC_{90}$ involved ordering the MIC values in ascending order and visually confirming which concentration corresponded to 50% and 90% inhibition, respectively.

Among the drugs tested in this study, cefoxitin was the only drug with established breakpoints for rapidly growing mycobacteria according to the CLSI guidelines (28). For the remaining drugs, we referred to the susceptibility of the same family drugs, imipenem and meropenem (MIC ≤ 4 µg/mL is susceptible), in addition to that of cefoxitin (MIC ≤ 16 µg/mL is susceptible).

## Statistical analysis

Statistical analysis was performed using the EZR version 1.62 (Saitama Medical Center, Jichi Medical University, Japan). As the data presented in this study were not normally distributed, nonparametric methods were used for analysis. Continuous variables for paired samples were compared using the Wilcoxon signed-rank test. All *P*-values were two-sided, and statistical significance was set at *P* < 0.05.

## RESULTS

### Identification of subspecies, erm(41) sequences, and colony morphotypes

All 43 isolates were identified as *M. abscessus*. Among these 43 *M. abscessus* isolates, 41 (95%) were associated with the T28 sequevar, while 2 (5%) were associated with the T28C sequevars. Among the tested isolates, 20 (18 clinical isolates and 2 QC strains) exhibited the rough morphotype, and 25 exhibited the smooth morphotype. None of the colonies were of indeterminate morphotype.

### Evaluation of quality control

QC was performed using *M. peregrinum* ATCC 700686. The MIC of cefoxitin in QC was 8 µg/mL (Table 1), within the QC range (4–32 µg/mL) mentioned in CLSI M24S (28).

### Activity of β-lactams with or without xeruborbactam against the 43 clinical isolates

The MIC values for individual β-lactams with or without 4 µg/mL xeruborbactam against the 43 *M. abscessus* clinical isolates were evaluated. Results showed that the MIC values tended to be higher for the rough morphotypes than for the smooth morphotypes (by one dilution; Table 2). For one of the clinical isolates (No. 7), adequate growth was not observed within the initial 5 days; therefore, we continued the incubation and determined the MIC value on day 10. The MIC distributions of individual β-lactams with or without 4 µg/mL xeruborbactam are shown in Fig. 1, and their MIC range, $MIC_{50}$, and $MIC_{90}$ values are presented in Table 2. The $MIC_{90}$ value for individual β-lactams (except for cefoxitin) exceeded 128 µg/mL, which was not expected for antimicrobial activity. The $MIC_{90}$ value of cefoxitin was slightly low at 32 µg/mL compared to that of other β-lactams, corresponding to intermediate susceptibility when referring to CLSI

**TABLE 1** MIC values of β-lactam antibiotics with or without 4 µg/mL xeruborbactam against *M. peregrinum* ATCC700686 and *M. abscessus* subsp. *absessus* ATCC 19977 in CAMHB

| Strain | β-Lactam antibiotic | MIC (µg/mL) | |
| --- | --- | --- | --- |
| | | Alone | With xeruborbactam |
| *M. peregrinum* ATCC700686 | Amoxicillin | 256 | 16 |
| | Tebipenem | 2 | 4 |
| | Cefdinir | 128 | >256 |
| | Cefuroxime | >256 | >256 |
| | Cefoxitin | 8 | 8 |
| *M. abscessus* ATCC19977 | Amoxicillin | >256 | 8 |
| | Tebipenem | 64 | 4 |
| | Cefdinir | 128 | 16 |
| | Cefuroxime | 256 | 32 |
| | Cefoxitin | 16 | 32 |

M24S (28). As with the *M. abscessus* ATCC 19977 strain, xeruborbactam lowered the MIC values of all β-lactams, except for cefoxitin, against the 43 clinical isolates. The MIC range shifted significantly to lower values in the following order: tebipenem > amoxicillin > cefuroxime > cefdinir. The lowest $MIC_{90}$ value was obtained from tebipenem with xeruborbactam; 1/43 isolates (2%) had an MIC of 2 µg/mL, and 42/43 isolates (98%) had an MIC of 4 µg/mL. The Wilcoxon signed-rank test indicated significant differences for amoxicillin ($P < 0.001$), tebipenem ($P < 0.001$), cefuroxime ($P < 0.001$), and cefdinir ($P < 0.001$) but not for cefoxitin ($P = 0.124$). The $MIC_{50}$ and $MIC_{90}$ values of cefoxitin without xeruborbactam were 16 µg/mL and 32 µg/mL, respectively, and they did not change upon the addition of xeruborbactam.

## DISCUSSION

This study demonstrated that xeruborbactam restored the activity of certain β-lactams against *M. abscessus*, which is most notably seen for tebipenem. Xeruborbactam lowered the $MIC_{90}$ values of tebipenem, amoxicillin, cefuroxime, and cefdinir by 5, ≥4, 3, and 3 dilutions, respectively.

Diazabicyclooctane β-lactamase (avibactam [10–12], relebactam [13, 14], durlobactam [15], zidebactam [16], and nacubactam [13, 16]) and cyclic boronic acid β-lactamase (vaborbactam [14]) inhibitors reportedly inhibit $Bla_{Mab}$. The use of combination therapy involving β-lactams along with novel β-lactamase inhibitors against MABC has primarily been investigated *in vitro*. Only two *in vivo* studies (29, 30) and one case report on using imipenem-relebactam for *M. abscessus* skin and soft tissue infection are available (31).

**TABLE 2** MIC values of β-lactam antibiotics with or without 4 µg/mL xeruborbactam against 43 *M. abscessus* subsp. *abscessus* clinical strains divided according to colony morphotype in CAMHB[a]

| Antibiotic(s) | Range | MIC (µg/mL) | | | | | |
| --- | --- | --- | --- | --- | --- | --- | --- |
| | | $MIC_{50}$ | | | $MIC_{90}$ | | |
| | | All | Smooth | Rough | All | Smooth | Rough |
| AMX | 64–>256 | >256 | >256 | >256 | >256 | >256 | >256 |
| AMX with XER | 4–64 | 8 | 8 | 8 | 32 | 16 | 32 |
| TBP | 4–>256 | 64 | 64 | 128 | 128 | 128 | 128 |
| TBP with XER | 2–4 | 4 | 4 | 4 | 4 | 4 | 4 |
| CDR | 8–>256 | 64 | 64 | 64 | 128 | 128 | 128 |
| CDR with XER | 8–32 | 8 | 8 | 16 | 16 | 16 | 16 |
| CXM | 8–>256 | 128 | 128 | 256 | 256 | >256 | 256 |
| CXM with XER | 8–64 | 16 | 8 | 16 | 32 | 16 | 32 |
| FOX | 16–64 | 16 | 16 | 32 | 32 | 64 | 32 |
| FOX with XER | 16–64 | 16 | 16 | 32 | 32 | 32 | 32 |

[a]AMX, amoxicillin; CDR, cefdinir; CXM, cefuroxime; FOX, cefoxitin; TBP, tebipenem; XER, xeruborbactam.

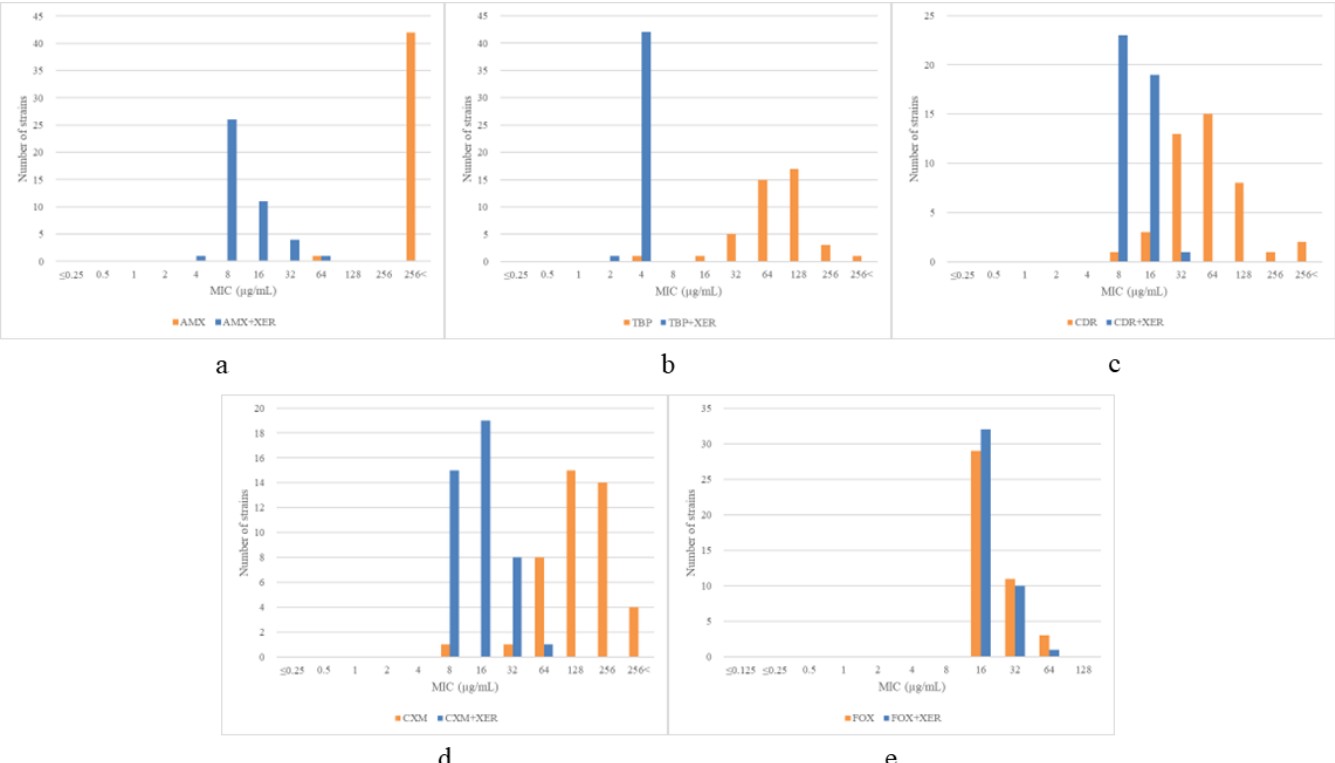

**FIG 1** MIC distributions of β-lactam antibiotics with or without 4 µg/mL xeruborbactam against the 43 *M. abscessus* subsp. *abscessus* clinical strains. (a) AMX; (b) TBP; (c) CDR; (d) CXM; (e) FOX. AMX, amoxicillin; CDR, cefdinir; CXM, cefuroxime; FOX, cefoxitin; TBP, tebipenem; XER, xeruborbactam.

To our knowledge, this was the first described case of the successful treatment of an *M. abscessus* infection using a novel β-lactamase inhibitor.

Previous reports indicated that novel β-lactamase inhibitors usually lowered the MIC values of tebipenem, cefuroxime, and cefdinir by 1–5, 2–4 or more, and 1–2 dilutions, respectively (10, 11, 13–16, 21). The decrease in the MIC values of amoxicillin varied according to the report, sometimes decreasing by approximately three dilutions or not at all. In this study, the MIC values of tebipenem, cefuroxime, cefdinir, and amoxicillin were reduced when these drugs were used in combination with xeruborbactam, and tebipenem demonstrated the most significant reduction, which were predictable results. In Fig. 1a and b, a well-defined Gaussian distribution is observed for tebipenem. Interestingly, when xeruborbactam was added, all MIC values converged around 4 µg/mL. Kaushik et al. (12) suggested that this concentration might represent a lower limit for the combination of carbapenem and avibactam, potentially explaining the observed convergence of MIC values.

Among the drugs used in this study, only cefoxitin had established breakpoints for MABC as per the CLSI guidelines (28). Therefore, to circumvent this issue, the susceptibilities of the same family drugs, imipenem and meropenem (MIC ≤ 4 µg/mL is susceptible), in addition to those of cefoxitin (MIC ≤ 16 µg/ml is susceptible) were used as reference. In such cases, comparison of individual MICs with peak plasma concentration, half-life, and plasma protein binding may be useful clinically (tebipenem: peak plasma concentration 14.1 µg/mL, half-life 1 h, and plasma protein binding 50% [32, 33]; imipenem: peak plasma concentration 30–35 µg/mL, half-life 1 h, and plasma protein binding 20% [34]; meropenem: peak plasma concentration 26 µg/mL, half-life 1 h, and plasma protein binding 2% [34]; cefuroxime: peak plasma concentration 2.1–13.6 µg/mL, half-life 2.2–3 h, and plasma protein binding 33%–50% [35]; cefdinir: peak plasma concentration 1.6–2.87 µg/mL, 1.5–1.7 h, and plasma protein binding 60%–73% [36]; cefoxitin: peak plasma concentration 42 µg/mL, 49 min, and plasma protein binding 50% [37]). With

reference to the above susceptibilities, the number of susceptible isolates increased from 2.3% (1 of 43) to 79.1% (34 of 43) for cefuroxime, from 9.3% (4 of 43) to 97.7% (42 of 43) for cefdinir, and from 2.3% (1 of 43) to 100% (43 of 43) for tebipenem with xeruborbactam. All isolates became susceptible when tebipenem was used with xeruborbactam. A systematic review of the combination therapy with carbapenems and novel β-lactamase inhibitors showed that the combination of tebipenem with avibactam against MABC was effective (21). Previous studies on multiple MABC clinical isolates reported that adding avibactam to tebipenem reduced the $MIC_{50}$ from 256 to 16 µg/mL (38). Although our study included a smaller number of isolates, the addition of xeruborbactam to tebipenem also led to a significant reduction in $MIC_{50}$, from 64 to 4 µg/mL. These findings suggest that the combination of tebipenem and xeruborbactam could be a promising addition to the treatment options for MABC infections.

The combinations ceftazidime-avibactam, sulbactam-durlobactam, imipenem-relebactam, and meropenem-vaborbactam are commercially available in the United States. However, ceftazidime is not active against MABC, even in combination with avibactam. Additionally, it is not effective against *M. abscessus* in which the gene encoding $Bla_{Mab}$ is deleted (9, 10, 39). Although sulbactam inhibits penicillin-binding protein, it is potent at only high concentrations (39, 40). Thus, the combination of β-lactam with ceftazidime-avibactam or sulbactam-durlobactam is needed to ensure efficient $Bla_{Mab}$ inhibition. However, this introduces the risk of ceftazidime- or sulbactam-related adverse events. Using ceftazidime-avibactam or sulbactam-durlobactam for MABC therapy would be inappropriate. In contrast, imipenem-relebactam and meropenem-vaborbactam may be active against MABC (14); nevertheless, both drugs need intravenous administration, and using them as outpatient treatment in the continuation phase is complicated.

Xeruborbactam is being developed as an injectable form with meropenem and an oral form with ceftibuten (41). Although it is unknown whether these combinations are effective against MABC, these can be administered both intravenously and orally and would likely be suitable for treating MABC, particularly as an all-oral regimen for outpatient treatment in the continuation phase. Furthermore, the combination of dual β-lactams and novel β-lactamase inhibitors has recently been reported to be more effective against MABC (27). Evaluation of the combined effects of meropenem-xeruborbactam or ceftibuten-xeruborbactam with other β-lactams for real-world clinical use is essential.

The primary adverse events of tebipenem, cefuroxime, and cefdinir are diarrhea, nausea, vomiting, and headache; however, most of them are mild or moderate in severity and non-treatment-limiting (35, 36, 42). Xeruborbactam has completed phase 1 clinical trials (NCT04380207, NCT04578873) in healthy adults administered through the intravenous or oral route. In these reports, xeruborbactam was safe and well tolerated at exposures that exceeded non-clinical PK-PD targets (24, 25). Because combining these β-lactams and xeruborbactam may cause unexpected adverse events or increase their incidence rates, further investigation is needed. Considering the incidence rate of the adverse events of currently recommended therapies, studying this combination is crucial.

As previously observed with avibactam, relebactam, nacubactam, zidebactam, and vaborbactam (10, 14, 16), the MIC value of cefoxitin was not affected by the presence of xeruborbactam ($MIC_{50}$ of 16 µg/mL and $MIC_{90}$ of 32 µg/mL with or without xeruborbactam). Soroka et al. (9) and Dubée et al. (43) reported that cefoxitin was slowly hydrolyzed by $Bla_{Mab}$, revealing its stability in the presence of $Bla_{Mab}$.

We analyzed the differences in the MIC values by colony morphotype. The MIC values tended to be higher for rough morphotypes than for smooth morphotypes by one dilution. The difference between the smooth and rough morphotypes is based on the glycopeptidolipids (GPLs) in the mycobacterial cell wall; smooth morphotypes possess GPL, whereas rough morphotypes do not. GPL conveys hydrophilicity, and the absence of GPL increases bacterial hydrophobicity (44). All antibiotics used in this study were hydrophilic, which could account for the low susceptibility of the rough morphotypes. Furthermore, the absence of GPL facilitates bacterial aggregation, clumping, and cording

(44). This may have prevented the drug from reaching the bacteria. The same results were reported in previous studies (13, 45). Although there are *in vitro* studies, the recent retrospective multicenter cohort study focused on *in vitro* investigations revealed that patients infected with rough MABC colony morphotypes experienced poorer clinical outcomes, including cavitary pulmonary disease and a higher frequency of cough, compared to those with smooth isolates (17). Consequently, when culturing MABC, it may be prudent to consider not only MIC values but also the colony morphotypes.

This study has some limitations. First, because the clinical isolates used in this study were obtained from a single hospital, the number of isolates was relatively small and may differ from those in other locations. Second, only *M. abscessus* isolates were evaluated in this study. A previous study showed that the proportion of patients with sustained sputum conversion rate without relapse was 23% for *M. abscessus* and 84% for *M. massiliense*, owing to differences in macrolide susceptibility (6, 46). Furthermore, obtaining *M. bollettii* strains was challenging because *M. bollettii* pulmonary disease cases are rare in Japan (47). This is why we focused on *M. abscessus* strains in this study. Third, the results of *in vitro* studies might not be applicable to clinical cases because MIC values reflect results in the predominant strains and likely do not reflect results in the subpopulation. However, we confirmed the repeatability and reproducibility using each β-lactam alone on the QC strain. Further *in vitro* and *in vivo* studies are needed to evaluate whether combining xeruborbactam with cefuroxime, cefdinir, and tebipenem is effective for treating other clinical strains, the extent of their bactericidal action and therapeutic efficacy, and the incidence of adverse events.

To the best of our knowledge, this study is the first to evaluate the effect of combining xeruborbactam with β-lactams against *M. abscessus*, and the findings of this study are useful because of the need to develop novel and highly effective all-oral anti-MABC regimens. Xeruborbactam lowered the $MIC_{90}$ of tebipenem, amoxicillin, cefuroxime, and cefdinir by 5, ≥4, 3, and 3 dilutions, respectively. The lowest $MIC_{90}$ value was obtained from tebipenem with xeruborbactam, indicating that all isolates possibly became susceptible when using this combination. The combination of tebipenem and xeruborbactam could be considered effective for total oral regimens used in the outpatient treatment of *M. abscessus* pulmonary disease.

## ACKNOWLEDGMENTS

We are grateful to all the hospital and laboratory personnel who supported this research.

This research was supported by the Research Program on Emerging and Re-emerging Infectious Disease from Japan Agency for Medical Research and development, AMED (grant number: JP23fk0108673).

I.Y.: investigation, data curation, formal analysis, and writing—original draft. A.A.: investigation. K.F.: formal analysis. T.A.: writing—review and editing. K.K.: writing—review and editing. Y.M.: writing—review and editing. Y.I.: writing—review and editing. K.C.: writing—review and editing. Y.M.: writing—review and editing. H.Y.: writing—review and editing. A.T.: writing—review and editing. K.K.: writing—review and editing. S.M.: conceptualization, methodology, investigation, writing—original draft, and writing—review and editing. All authors critically read and commented on the manuscript.

## AUTHOR AFFILIATIONS

[1]Department of Mycobacterium Reference and Research, Research Institute of Tuberculosis, Japan Anti-Tuberculosis Association, Tokyo, Japan
[2]Respiratory Medicine and Infectious Diseases, Oita University Faculty of Medicine, Oita, Japan
[3]Department of Basic Mycobacteriosis, Nagasaki University Graduate School of Biomedical Sciences, Nagasaki, Japan

## AUTHOR ORCIDs

Izumi Yamatani http://orcid.org/0009-0006-4535-8198
Keiji Fujiwara https://orcid.org/0000-0002-1139-1604
Yuta Morishige https://orcid.org/0000-0002-3818-5288
Yoshiro Murase https://orcid.org/0000-0003-0240-8352
Kosaku Komiya https://orcid.org/0000-0002-8679-1661
Satoshi Mitarai http://orcid.org/0000-0002-2288-077X

## FUNDING

| Funder | Grant(s) | Author(s) |
|---|---|---|
| Japan Agency for Medical Research and Development (AMED) | JP23fk0108673 | Satoshi Mitarai |

## AUTHOR CONTRIBUTIONS

Izumi Yamatani, Conceptualization, Data curation, Formal analysis, Investigation, Methodology, Project administration, Resources, Validation, Visualization, Writing – original draft, Writing – review and editing | Akio Aono, Investigation, Validation, Writing – review and editing | Keiji Fujiwara, Formal analysis, Writing – review and editing | Takahiro Asami, Writing – review and editing | Keisuke Kamada, Writing – review and editing | Yuta Morishige, Writing – review and editing | Yuriko Igarashi, Writing – review and editing | Kinuyo Chikamatsu, Writing – review and editing | Yoshiro Murase, Writing – review and editing | Hiroyuki Yamada, Writing – review and editing | Akiko Takaki, Writing – review and editing | Kosaku Komiya, Writing – review and editing | Satoshi Mitarai, Conceptualization, Funding acquisition, Methodology, Project administration, Resources, Supervision, Writing – review and editing

## ADDITIONAL FILES

The following material is available online.

### Open Peer Review

**PEER REVIEW HISTORY (review-history.pdf).** An accounting of the reviewer comments and feedback.

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
