## [Reviewer comments · Microbiology Spectrum]

Microbiology Spectrum

***In vitro* effects of the new oral β -lactamase inhibitor xeruborbactam in combination with oral β -lactams against clinical *Mycobacterium abscessus* isolates**

Izumi Yamatani, Akio Aono, Keiji Fujiwara, Takahiro Asami, Keisuke Kamada, Yuta Morishige, Yuriko Igarashi, Kinuyo Chikamatsu, Yoshiro Murase, Hiroyuki Yamada, Akiko Takaki, Kosaku Komiya, and Satoshi Mitarai

Corresponding Author(s): Izumi Yamatani, Koeki Zaidan Hojin Kekkaku Yobokai Kekkaku Kenkyujo Kosankinbu

Review Timeline:

Submission Date:	January 21, 2024
Editorial Decision:	March 11, 2024
Revision Received:	April 17, 2024
Editorial Decision:	April 23, 2024
Revision Received:	April 29, 2024
Accepted:	May 2, 2024

Editor: Olivier Neyrolles

Reviewer(s): Disclosure of reviewer identity is with reference to reviewer comments included in decision letter(s). The following individuals involved in review of your submission have agreed to reveal their identity: Lina Davies Forsman (Reviewer #1); Emmanuelle Cambau (Reviewer #2)

Transaction Report:

DOI: <https://doi.org/10.1128/spectrum.00084-24>

Re: Spectrum00084-24 (*In vitro* Effects of the New Oral β -lactamase Inhibitor Xeruboractam in Combination with Oral β -lactams Against *Mycobacterium abscessus* Clinical Isolates)

Dear Dr. Izumi Yamatani:

Thank you for the privilege of reviewing your work. Below you will find my comments, instructions from the Spectrum editorial office, and the reviewer comments.

As recommended by Reviewer #2, I suggest you shorten the manuscript and merge the Results and Discussion sections.

Revision Guidelines

Sincerely,
Olivier Neyrolles
Editor
Microbiology Spectrum

Reviewer #2 (Comments for the Author):

Spectrum00084-24 Yamatani et al.

The authors determined Minimal inhibitory concentrations of various beta-lactams combined with a new inhibitor of beta-

lactamase against *Mycobacterium abscessus*.

It is data of interest since *M. abscessus* are intrinsically resistant to nearly all antimicrobials and the need of novel antimicrobials or novel antibiotic combination is highly required.

Beta-lactams are on the first line list of antimicrobial therapy for *M. abscessus* infections, particularly imipenem and ceftazidime. Since *M. abscessus* produces a beta-lactamase, several combinations with beta-lactamase inhibitors were tested, showing a decrease of some MIC dilutions.

This study tested a novel inhibitor of beta-lactamase, xeruborbactam, which belongs to the cyclic boronate acid beta-lactamase inhibitors and has the interesting pharmacokinetic enabling an oral administration.

Major comments and questions:

- It is very interesting to have these data about a new combination of penem and inhibitor of beta lactamase that could be delivered orally to patients infected with *M. abscessus*. However, the paper contains limited data (MIC testing on 44 isolates for 5 beta-lactams combined with xeruborbactam), and consequently can be shorten into a note of a brief report.
- MIC protocol and description of the *M. abscessus* isolates results are according to the reference. However, we would expect to see results of repeatability and reproducibility at least on the reference strain.
- Results show interesting data with low MIC of the combination xeruborbactam and terbipenem.
- Bibliography is consistent with the topic.

Minor comments:

1. abstract : the results for ceftazidime are missing

2. MIC determination protocol:

a. page 7, lines 118-121. Is the xeruborbactam water soluble, or is it dissolved in a specific solvent, e.g. DMSO?

b. Page 8, line 134: waiting for 14 days could increase the MIC by degrading the inhibitor or the beta-lactam, which is often observed for penems. For how many isolates (for one isolate, the reading was done at D10), this extended incubation was used? If it is zero, better to suppress this sentence.

3. Results:

a. Tables: results shown in table 3 can be combined with those of the table 2.

b. Figure: how do you explain that there is a nice Gauss distribution observed for terbipenem on the figure 1b, and the all MIC results are concentrated on the 4 µg/mL value when the xeruborbactam is added?

4. Discussion:

a. Page 12, line 208. Since xeruborbactam will be commercialized with ceftibuten as an oral form, it could have been worthwhile to test this combination.

b. The authors showed that MIC on Rough isolates are higher than on the Smooth isolates? Page 13, Lines 230-231, the authors explained that the absence of GPL enhance the permeability cell wall. So consequently, the MIC should be lower for the R isolates? Do they have hypotheses for this unexpected result?

Re: *In vitro* effects of the new oral β -lactamase inhibitor xeruborbactam in combination with oral β -lactams against *Mycobacterium abscessus* clinical isolates

The authors have studied the effect of five oral β -lactams against *Mycobacterium abscessus* isolates. There is an urgent need to improve the treatment of *M.abscessus* and oral treatment options are needed. The study is therefore of interest for the readers of Spectrum Microbiology.

The manuscript is clear, well-written and a joy to read, apart from some small mistakes. I do have some suggestions to improve the manuscript, including readability.

Major comments:

The introduction is rather lengthy, and parts of the text can be moved to the discussion part.

Line 46. How were the critical concentrations decided? There is no CC established for tebipenem so this needs to be noted in the methods section and possibly discussed in the discussion section.

Line 98. What is this Ogawa medium? Used outside Japan? Consists of?

Add in M& M section about how colony morphology was determined. Any indeterminate? Add this as a secondary aim as it is an important finding and deserves more discussion about. Consider referring to previously published studies on the subject:

Hedin W, Fröberg G, Fredman K, Chryssanthou E, Selmeryd I, Gillman A, Orsini L, Runold M, Jönsson B, Schön T, Davies Forsman L. A Rough Colony Morphology of *Mycobacterium abscessus* Is Associated With Cavitory Pulmonary Disease and Poor Clinical Outcome. *J Infect Dis.* 2023 Mar 28;227(6):820-827. doi: 10.1093/infdis/jiad007. PMID: 36637124; PMCID: PMC10043986.

Minor comments:

In general, up to the authors of course, but the readability of the manuscript would be improved if abbreviations are kept to a minimum. I would not abbreviate the B-lactam antibiotics as all readers are not aware of these abbreviations.

Suggest rewording the title:

In vitro effect of the new oral β -lactamase inhibitor xeruborbactam in combination with oral β -lactams against clinical *Mycobacterium abscessus* isolates

Line 31. Rephrase –. Almost all isolates had an MIC of 4 mg/L, whereas one isolate had an MIC of 2 mg/L.

Line 39. Change “it will be needed” to “There is a need to”....

Line 42. “which is inhibited by cyclic boronate acid β -lactamase inhibitors.” Add: ..., amongst other. As the sentence reads now you can think that it is only cyclic boronate that inhibits it.

Line 45. B-lactam antibiotic (add antibiotic)

Line 54. MABC instead of MABS (MAB complex is more commonly used)

Line 64. Add: The (The T28C)

Line 81. Plural form – β -lactams

84. This is the first case of ... Rephrase – you get the impression that this study will describe this as you use the present tense. Use past tense to describe previous studies. Make it clearer that you are referring to the case report mentioned before.

Line 123. What is the half-life of XER? Is it stable during long incubation times? At least discuss this as a possible limitation. This has been discussed previously with clavulanic acid when MER-CLA has been tested.

Line 157. QC – write out (quality control) at least in the subheading

Line 168. Rephrase “slightly low”. Very subjective. Compare to something or state the MIC

Line 171. Rephrase – not clear enough

Line 176. Change to “upon the addition of...”

Line 180. Suggest rephrasing to: This study demonstrated that XER restored the activity of certain β -lactams against *M. abscessus*, most notably seen for tebipenem.

Line 198 line. I would recommend comparing your MICs with previous studies. Reference the systematic review is ok but also add info and reference specific studies, and discuss the difference in results (“the addition of avibactam to tebipenem showed a more pronounced reduction from 256 to 16 mg/L”

Fröberg G, Ahmed A, Chryssanthou E, Davies Forsman L. 2023. The in vitro effect of new combinations of carbapenem- β -lactamase inhibitors for *Mycobacterium abscessus*. *Antimicrob Agents Chemother* 67:e00528-23.

<https://doi.org/10.1128/aac.00528-23>

Line 200. Add “Thus” or explain why as the sentence is not clear enough now.

Line 204. Change to “may be” instead of “should be” as the latter is an opinion.

Spectrum00084-24

Response to Reviewers

We thank the Reviewers for their useful comments that helped us to make significant improvements to our manuscript, and we greatly appreciate the time and effort that they spent on our behalf.

Reviewer #1

1. The authors have studied the effect of five oral β -lactams against *Mycobacterium abscessus* isolates. There is an urgent need to improve the treatment of *M. abscessus* and oral treatment options are needed. The study is therefore of interest for the readers of Spectrum Microbiology.

The manuscript is clear, well-written and a joy to read, apart from some small mistakes. I do have some suggestions to improve the manuscript, including readability.

Authors' response:

Thank you for your helpful comments.

2. The introduction is rather lengthy, and parts of the text can be moved to the discussion part.

Authors' response:

Thank you for your suggestions. We have moved the text about the details of novel β -lactamase inhibitors, *in vivo* studies, and a case report to the Discussion section.

3. Line 46 "the number of susceptible isolates increased from 1/43 (2%) to 43/43 (100%) for tebipenem with xeruborbactam". How were the critical concentrations decided? There is no CC established for tebipenem so this needs to be noted in the methods section and possibly discussed in the discussion section.

Authors' response:

Thank you for your suggestion. We have added in the Methods and Discussion sections that there was no CC established for tebipenem, cefuroxime, cefdinir, and amoxicillin. We referred to the susceptibility of the same family drug, imipenem and meropenem (MIC ≤ 4 $\mu\text{g/mL}$ is susceptible), and ceftazidime (MIC ≤ 16 $\mu\text{g/mL}$ is susceptible). At that time, the comparison of individual MICs with peak plasma concentration, half-life, and plasma protein binding may be useful clinically; thus, we have added that information about each β -lactams.

4. Line 98 "2% Ogawa medium (Kyokuto Pharmaceutical Industrial, Tokyo, Japan)". What is this Ogawa medium? Used outside Japan? Consists of?

Authors' response:

Thank you for pointing this out. The 2% Ogawa medium is mainly used in Japan and some Asian countries, instead of the Löwenstein-Jensen medium, which is an egg-based medium and consists of potassium dihydrogen phosphate, sodium glutamate, magnesium citrate, soluble starch, sodium pyruvate, glycerol, 2% malachite green, and homogenized whole egg. We have added this information in the Methods section.

5. Add in M& M section about how colony morphology was determined. Any indeterminate? Add this as a secondary aim as it is an important finding and deserves more discussion about. Consider referring to previously published studies on the subject: Hedin W, et al. J Infect Dis. 2023 Mar 28;227(6):820-827.

Authors' response:

Thank you for your suggestion. We have added details on how colony morphology was determined in the Methods section, and there was no indeterminate colony morphotype in the Results section. We have discussed the clinical significance of colony morphology and set the secondary aim to evaluate the differences in the effect of colony morphology.

6. In general, up to the authors of course, but the readability of the manuscript would be improved if abbreviations are kept to a minimum. I would not abbreviate the B-lactam antibiotics as all readers are not aware of these abbreviations.

Authors' response:

Thank you for your suggestion. β -lactam has been spelled out throughout the text accordingly.

7. Suggest rewording the title:
In vitro effect of the new oral β -lactamase inhibitor xeruborbactam in combination with oral β -lactams against clinical *Mycobacterium abscessus* isolates

Authors' response:

Thank you for your comment. The title has been revised according to your suggestion.

8. Line 31. Rephrase –. Almost all isolates had an MIC of 4 mg/L, whereas one isolate had an MIC of 2 mg/L.

Authors' response:

Thank you for your suggestion. We have revised according to your suggestion.

9. Line 39. Change “it will be needed” to “There is a need to”....

Authors' response:

Thank you for your suggestion. We have revised it according to your suggestion.

10. Line 42. “which is inhibited by cyclic boronate acid β -lactamase inhibitors.” Add: ..., amongst other. As the sentence reads now you can think that it is only cyclic boronate that inhibits it.

Authors' response:

Thank you for your suggestion. We have revised it according to your suggestion.

11. Line 45. B-lactam antibiotic (add antibiotic)

Authors' response:

Thank you for your correction. We have revised it according to your suggestion.

12. Line 54. MABC instead of MABS (MAB complex is more commonly used)

Authors' response:

Thank you for your suggestion. We have revised the term according to your suggestion.

13. Line 64. Add: The (The T28C)

Authors' response:

Thank you for pointing this out. We have revised it according to your suggestion.

14. Line 81. Plural form – β -lactams

Authors' response:

Thank you for your correction. We have revised it according to your suggestion.

15. Line 84. This is the first case of ... Rephrase – you get the impression that this study will describe this as you use the present tense. Use past tense to describe previous studies. Make it clearer that you are referring to the case report mentioned before.

Authors' response:

Thank you for your suggestion. We have revised it according to your suggestion.

16. Line 123. What is the half-life of XER? Is it stable during long incubation times? At least discuss this as a possible limitation. This has been discussed previously with clavulanic acid when MER-CLA has been tested.

Authors' response:

The half-life of xeruborbactam is estimated to be approximately 30 h, while the half-life of avibactam is approximately 2 h. Although the stability of xeruborbactam during long-time incubation is still unknown, avibactam is reported to be stable. Therefore, we have considered that xeruborbactam was probably stable during incubation.

17. Line 157. QC – write out (quality control) at least in the subheading.

Authors' response:

Thank you for your suggestion. We have revised it according to your suggestion.

18. Line 168. Rephrase “slightly low”. Very subjective. Compare to something or state the MIC.

Authors' response:

Thank you for pointing this out. We have made the comparisons with other β -lactams accordingly.

19. Line 171. Rephrase – not clear enough

Authors' response:

We have added the sentence that the MIC range shifted “significantly to lower values” in the order of tebipenem, amoxicillin, cefuroxime, and cefdinir.

20. Line 176. Change to “upon the addition of...”

Authors' response:

Thank you for your suggestion. We have revised it according to your suggestion.

21. Line 180. Suggest rephrasing to: This study demonstrated that XER restored the activity of certain β -lactams against *M. abscessus*, most notably seen for tebipenem.

Authors' response:

Thank you for your suggestion. We have revised it accordingly.

22. Line 198. I would recommend comparing your MICs with previous studies. References the systematic review is ok but also add info and reference specific studies, and discuss the difference in results (“ the addition of avibactam to tebipenem showed a more pronounced reduction from 256 to 16 mg/L”

Fröberg G, et al. 2023. Antimicrob Agents Chemother 67:e00528-23.

Authors' response:

Thank you for your suggestions. We have added the information about the specific study you indicated and discussed the differences in the Discussion section.

23. Line 200. Add ”Thus” or explain why as the sentence is not clear enough now.

Authors' response:

Thank you for your suggestion. We have revised it according to your suggestion.

24. Line 204. Change to “may be” instead of “should be” as the latter is an opinion.

Authors' response:

Thank you for your suggestion. We have revised it accordingly.

Reviewer #2

1. The authors determined Minimal inhibitory concentrations of various beta-lactams combined with a new inhibitor of beta-lactamase against *Mycobacterium abscessus*. It is data of interest since *M. abscessus* are intrinsically resistant to nearly all antimicrobials and the need of novel antimicrobials or novel antibiotic combination is highly required.
Beta-lactams are on the first line list of antimicrobial therapy for *M. abscessus* infections, particularly imipenem and ceftazidime. Since *M. abscessus* produces a beta-lactamase, several combinations with beta-lactamase inhibitors were tested, showing a decrease of some MIC dilutions.
This study tested a novel inhibitor of beta-lactamase, xeruborbactam, which belongs to the cyclic boronate acid beta-lactamase inhibitors and has the interesting pharmacokinetic enabling an oral administration.

Authors' response:

Thank you for your helpful comments.

2. It is very interesting to have these data about a new combination of penem and inhibitor of beta lactamase that could be delivered orally to patients infected with *M. abscessus*. However, the paper contains limited data (MIC testing on 44 isolates for 5 beta-lactams combined with xeruborbactam), and consequently can be shorten into a note of a brief report.

Authors' response:

Thank you for your helpful comments. However, because this report has two tables and one figure, and we have added some discussions according to the reviewers' comments, it was difficult to shorten the text into a brief report. We are very much grateful if you kindly understand our considerations.

3. MIC protocol and description of the *M. abscessus* isolates results are according to the reference. However, we would expect to see results of repeatability and reproducibility at least on the reference strain.

Authors' response:

Thank you for your suggestion. However, we are running out of xeruborbactam and are unable to do the re-test immediately. It may take one more month to get xeruborbactam, in which case we will likely not be able to return our manuscript within 60 days. It is a last resort, but we confirmed the repeatability and reproducibility using each β -lactam alone at least on the QC strain.

4. Results show interesting data with low MIC of the combination xeruborbactam and terbipenem.

Authors' response:

Thank you for your helpful comments.

5. Bibliography is consistent with the topic.

Authors' response:

Thank you for your helpful comments.

6. abstract: the results for cefoxitin are missing

Authors' response:

Thank you for pointing this out. We have added the results of cefoxitin in the Abstract.

7. page 7, lines 118-121. Is the xeruborbactam water soluble, or is it dissolved in a specific solvent, e.g. DMSO?

Authors' response:

According to the manufacturer's instructions, xeruborbactam can be dissolved in DMSO, and there is no data available about water solubility. We have added that information in the Methods section. In this study, we dissolved xeruborbactam and each β -lactam directly in CAMHB. When measuring the MIC values of each β -lactam alone, a two-fold dilution series of the β -lactam stock solution was prepared using CAMHB. When measuring the MIC values of each β -lactam and xeruborbactam combination, a two-fold dilution series of the β -lactam stock solution was prepared using CAMHB with xeruborbactam.

8. Page 8, line 134: waiting for 14 days could increase the MIC by degrading the inhibitor or the beta-lactam, which is often observed for penems. For how many isolates (for one isolate, the reading was done at D10), this extended incubation was used? If it is zero, better to suppress this sentence.

Authors' response:

Extended incubation time beyond Day 5 was used for only one isolate (No. 7). We would like to keep this sentence.

9. Tables: results shown in table 3 can be combined with those of the table 2.

Authors' response:

We have combined Tables 2 and 3.

10. Figure: how do you explain that there is a nice Gauss distribution observed for terbipenem on the figure 1b, and the all MIC results are concentrated on the 4 $\mu\text{g/mL}$ value when the xeruborbactam is added?

Authors' response:

As previous study indicated that it appeared to be a lower limit for the MIC of carbapenem and avibactam combination, we have considered this as the reason for the

convergence of the MIC values. We have added the information in the Discussion section.

11. Page 12, line 208. Since xeruborbactam will be commercialized with ceftibuten as an oral form, it could have been worthwhile to test this combination.

Authors' response:

Thank you for your comment. As per your suggestions, we should have tested the combination of xeruborbactam and ceftibuten. When we started this study, we were not aware that xeruborbactam was being developed in combination with ceftibuten, and as we had run out of xeruborbactam, we could not test this combination this time, unfortunately.

12. The authors showed that MIC on Rough isolates are higher than on the Smooth isolates? Page 13, Lines 230-231, the authors explained that the absence of GPL enhance the permeability cell wall. So consequently, the MIC should be lower for the R isolates? Do they have hypotheses for this unexpected result?

Authors' response:

The absence of GPL may enhance the permeability of the cell wall and the accumulation of the drug inside the bacteria. We have considered that this might prevent β -lactams from acting on the bacterial cell wall, leading to the low susceptibility of the rough morphotypes, and added that in the Discussion section.

Re: Spectrum00084-24R1 (*In vitro* effects of the new oral β -lactamase inhibitor xeruborbactam in combination with oral β -lactams against clinical *Mycobacterium abscessus* isolates)

Dear Dr. Izumi Yamatani:

Thank you for the privilege of reviewing your revised manuscript.

Before I can proceed with formal acceptance, I would like you to address the remaining comments raised by Reviewer #1.

Revision Guidelines

Sincerely,
Olivier Neyrolles
Editor
Microbiology Spectrum

Reviewer #1 (Comments for the Author):

Thanks for a nicely revised manuscript and overall clear answers.

1. I suggest to rephrase this statement to make it less bold, as this is solely *in vitro* data:

"We showed that the tebipenem-xeruborbactam combination could be an 53 effective oral regimen that benefits outpatient

treatment of M. abscessus pulmonary disease."

Suggest to write "may be of interest to explore further for" Or something similar

2. This remains unanswered and does not seem logical.

12. The authors showed that MIC on Rough isolates are higher than on the Smooth isolates? Page 13, Lines 230-231, the authors explained that the absence of GPL enhance the permeability cell wall. So consequently, the MIC should be lower for the R isolates? Do they have hypotheses for this unexpected result? Authors' response: The absence of GPL may enhance the permeability of the cell wall and the accumulation of the drug inside the bacteria. We have considered that this might prevent β -lactams from acting on the bacterial cell wall, leading to the low susceptibility of the rough morphotypes, and added that in the Discussion section.

3. Add as a limitation:

MIC protocol and description of the M. abscessus isolates results are according to the reference. However, we would expect to see results of repeatability and reproducibility at least on the reference strain. Authors' response: Thank you for your suggestion. However, we are running out of xeruborbactam and are unable to do the re-test immediately. It may take one more month to get xeruborbactam, in which case we will likely not be able to return our manuscript within 60 days. It is a last resort, but we confirmed the repeatability and reproducibility using each β -lactam alone at least on the QC strain.

Reviewer #2 (Comments for the Author):

Thank you for having considered the comments and questions

Spectrum00084-24R1

In vitro effects of the new oral β -lactamase inhibitor xeruboractam in combination with oral β lactams against clinical *Mycobacterium abscessus* isolates

Response to Reviewer #1

Thank you for reviewing our revised manuscript, and for your thoughtful comments, which helped us to improvement to our manuscript. We have revised the manuscript accordingly. The changes are shown in red font. We have provided a point-by-point response to your comments, below.

Reviewer #1

1. I suggest to rephrase this statement to make it less bold, as this is solely in vitro data: "We showed that the tebipenem-xeruboractam combination could be an effective oral regimen that benefits outpatient treatment of M. abscessus pulmonary disease." Suggest to write "may be of interest to explore further for" Or something similar.

Authors' response:

Thank you for your suggestion. We have revised the wording according to your suggestion. (Lines 52–53)

2. This remains unanswered and does not seem logical: The authors showed that MIC on Rough isolates are higher than on the Smooth isolates? Page 13, Lines 230-231, the authors explained that the absence of GPL enhance the permeability cell wall. So consequently, the MIC should be lower for the R isolates? Do they have hypotheses for this unexpected result?

Authors' response:

Thank you for your comments and questions. GPL convey hydrophilicity, and the absence of GPL increases bacterial hydrophobicity. All antibiotics used in this study were hydrophilic, which could account for the low susceptibility of the rough morphotypes.

Furthermore, the absence of GPL facilitates bacterial aggregation, clumping, and cording. This may have prevented the drug from reaching the bacteria. We have revised the Discussion section to provide this information. (Lines 312–315)

3. Add as a limitation: MIC protocol and description of the *M. abscessus* isolates results are according to the reference. However, we would expect to see results of repeatability and reproducibility at least on the reference strain.

Authors' response:

Thank you for your suggestions. We have added this information as a limitation in the Discussion section. (Lines 332–333)

Re: Spectrum00084-24R2 (*In vitro* effects of the new oral β -lactamase inhibitor xeruborbactam in combination with oral β -lactams against clinical *Mycobacterium abscessus* isolates)

Dear Dr. Izumi Yamatani:

Congratulations!

Your manuscript has been accepted, and I am forwarding it to the ASM production staff for publication. Your paper will first be checked to make sure all elements meet the technical requirements. ASM staff will contact you if anything needs to be revised before copyediting and production can begin. Otherwise, you will be notified when your proofs are ready to be viewed.

Sincerely,
Olivier Neyrolles
Editor
Microbiology Spectrum